# Dietary sodium chloride attenuates increased β-cell mass to cause glucose intolerance in mice under a high-fat diet

**Keigo Taki** **[1], Hiroshi Takagi[1] \*, Tomonori Hirose[1], Runan Sun[1], Hiroshi Yaginuma[1], Akira Mizoguchi[1], Tomoko Kobayashi[1], Mariko Sugiyama[1], Taku Tsunekawa[1], Takeshi Onoue[1], Daisuke Hagiwara[1], Yoshihiro Ito[1], Shintaro Iwama[1], Hidetaka Suga[1], Ryoichi Banno[1,2], Daisuke Sakano[3], Shoen Kume[3], Hiroshi Arima[1]**

1 Department of Endocrinology and Diabetes, Nagoya University Graduate School of Medicine, Showa-ku, Nagoya, Japan, 2 Research Center of Health, Physical Fitness and Sports, Nagoya University, Nagoya, Japan, 3 Department of Life Science and Technology, School of Life Science and Technology, Tokyo Institute of Technology, Midori-ku, Yokohama, Kanagawa, Japan

* htakagi@med.nagoya-u.ac.jp

## Abstract

Excessive sodium salt (NaCl) or fat intake is associated with a variety of increased health risks. However, whether excessive NaCl intake accompanied by a high-fat diet (HFD) affects glucose metabolism has not been elucidated. In this study, C57BL/6J male mice were fed a normal chow diet (NCD), a NCD plus high-NaCl diet (NCD plus NaCl), a HFD, or a HFD plus high-NaCl diet (HFD plus NaCl) for 30 weeks. No significant differences in body weight gain, insulin sensitivity, and glucose tolerance were observed between NCD-fed and NCD plus NaCl-fed mice. In contrast, body and liver weights were decreased, but the weight of epididymal white adipose tissue was increased in HFD plus NaCl-fed compared to HFD-fed mice. HFD plus NaCl-fed mice had lower plasma glucose levels in an insulin tolerance test, and showed higher plasma glucose and lower plasma insulin levels in an intraperitoneal glucose tolerance test compared to HFD-fed mice. The β-cell area and number of islets were decreased in HFD plus NaCl-fed compared to HFD-fed mice. Increased Ki67-positive β-cells, and increased expression levels of Ki67, CyclinB1, and CyclinD1 mRNA in islets were observed in HFD-fed but not HFD plus NaCl-fed mice when compared to NCD-fed mice. Our data suggest that excessive NaCl intake accompanied by a HFD exacerbates glucose intolerance, with impairment in insulin secretion caused by the attenuation of expansion of β-cell mass in the pancreas.

## Introduction

Excessive sodium chloride (NaCl) intake is associated with a variety of increased health risks, including hypertension [1,2], cardiovascular disease [3,4], and chronic kidney disease [5]. Although the World Health Organization has proposed 5 g/day as a maximum target for daily NaCl intake [6], salt intake exceeds recommended levels in most countries [7]. Excessive

**Funding:** This study was supported by JSPS KAKENHI Grant Numbers JP17K17803 and 19K20180 to HT; and Chukyo Longevity Foundation in 2019 to HT. The funders had no role in study design, data collection and analysis, decision to publish, or preparation of the manuscript.

**Competing interests:** The authors have declared that no competing interests exist.

intake of fat is known to be a cause of obesity [8], and is also associated with increased health risks. Recently, changes in dietary habits have increased the consumption of a Western-style diet, composed of processed foods that are high in NaCl and saturated fat. It is generally accepted that excessive consumption of NaCl and fat leads to poor health outcomes. However, a paucity of studies exists examining the effects of excess NaCl intake combined with fats.

Type 2 diabetes mellitus (DM) is a serious and commonly occurring chronic disease, often accompanied by micro- and macrovascular complications. The prevalence of increasing type 2 DM and its associated complications lower the quality of people's lives and impose enormous economic and social burdens [9,10]. In the pathogenesis of type 2 DM, impaired insulin sensitivity and insulin secretion were found to be major factors involved in the progression of diabetes. As is well known, an increase in the consumption of a high-fat diet (HFD) is associated with glucose intolerance [11]. In comparison, in rodent models, an association between an excess consumption of NaCl and insulin sensitivity has been controversial [12–14], and the effects of NaCl consumption on insulin secretion have not been clarified [15].

In the present study, we evaluated the influence of excessive NaCl on the glucose metabolism of HFD-fed mice in order to elucidate whether excessive consumption of NaCl accompanied by fat causes impaired glucose metabolism.

## Materials and methods

### Animals and diets

Seven-week-old C57BL/6J mice were obtained from Japan SLC (Shizuoka, Japan). Mice were maintained with a 12 h light/12 h dark cycle in a temperature-controlled barrier facility, and with free access to water and food. After 1 week acclimation, mice were divided into four groups and fed a normal chow diet (NCD; CE2; CLEA Japan, Tokyo, Japan; 0.78% NaCl, 24.9% protein, 4.6% fat, and 70.5% carbohydrate, n = 16), a NCD plus high salt diet (NCD plus NaCl; 4.0% NaCl, n = 16), a HFD (Test Diet 58Y1; PMI Nutrition International, Clayton, MO, USA; 0.38% NaCl, 18.3% protein, 60.9% fat, and 20.1% carbohydrate, n = 28), or a HFD plus high salt diet (HFD plus NaCl; 4.0% NaCl, n = 28) for 30 weeks. Body weight (BW) was monitored for the 30 weeks of feeding. After 30 weeks of a diet as outlined above, mice were sacrificed and subcutaneous white adipose tissue (WAT), mesenteric WAT, epididymal WAT, liver, gastrocnemius muscle, and soleus muscle were collected. All animal procedures were approved by the Animal Care and Use Committee of Nagoya University Graduate School of Medicine and performed in accordance with the institutional guidelines that conform to National Institutes of Health animal care guidelines. Mice were sacrificed using cervical dislocation by trained individuals, and all efforts were made to minimize suffering.

### Energy expenditure measurements

In another cohort, food intake and feed efficiency, which were calculated as grams of BW gained per grams of food consumed over a 3 day period [16], were evaluated at the age of 8 weeks (n = 15–16/group). Mice at this age were acclimated to the test cage for 24 h, and energy expenditure was measured at 5-min intervals for 24 h on the second day (Model Supermex; Muromachi Kikai, Tokyo, Japan) (n = 5/group). Oxygen consumption ($VO_2$) was measured using electrochemical and spectrophotometric sensors. Locomotor activity was measured simultaneously by infrared beam interruption (Model MK-500RQ/02; Muromachi Kikai, Tokyo, Japan) and reported as average counts per hour.

## Plasma biochemical analysis

Blood was collected by tail bleeding. Sera were separated by centrifugation at $6,000 \times g$. Serum levels of insulin were measured by enzyme-linked immunoassay (Morinaga Institute of Biochemical Science, Kanagawa, Japan). Blood glucose levels were measured with a glucometer (Glutestmint; Sanwa Kagaku Kenkyusho, Aichi, Japan). Serum total cholesterol, triglyceride and free fatty acid levels in the fasting state were measured with commercial kits (FUJIFILM Wako Pure Chemical Co., Osaka, Japan) on a Hitachi 7060 Automatic Analyzer (Hitachi High-Technologies Corporation, Tokyo, Japan).

## Insulin, glucose and pyruvate tolerance tests

An insulin tolerance test (ITT) was performed in mice 25 weeks after the start of diet interventions [17]. An intraperitoneal glucose tolerance test (GTT), which has been widely used to examine glucose tolerance, was performed 26 weeks after diet interventions, as described previously [17]. A pyruvate tolerance test (PTT) was carried out in mice after 27 weeks' feeding of a HFD or a HFD plus NaCl. For a PTT, mice were deprived of food for 16 h before the test, and blood was collected 0, 15, 30, 60, 90, and 120 min after the injection of pyruvate. Blood glucose was assayed in tail blood using a glucometer (Glutestmint). Additionally, the area under the curve (AUC) derived from each test was calculated. In the GTT, plasma insulin levels were measured 0 and 30 min after the injection of glucose. Measurements were taken at the onset of the light cycle between 9 and 10 am. The insulin dose used for intraperitoneal injections was 1.0 mU/g BW for mice on NCD and NCD plus NaCl, and 1.6 mU/g BW for those on HFD and HFD plus NaCl. The glucose dose used for intraperitoneal injections was 2 mg/g BW for mice on NCD and NCD plus NaCl, and 1 mg/g BW for those on HFD and HFD plus NaCl. The pyruvate dose used for intraperitoneal injections was 1.5 mg/g BW.

## Islet isolation

Mouse pancreatic islets were isolated using a collagenase digestion method as described previously [18] with some modifications. Briefly, mice fed NCD, HFD or HFD plus NaCl for 1 week were sacrificed (n = 20/NCD group, n = 13/HFD group, n = 12/HFD plus NaCl group), and the pancreas was exposed by laparotomy. The common bile duct was ligated at the ampulla of Vater and cannulated using a needle (Natsume Seisakusho, Tokyo, Japan). Thereafter, 4–5 mL solution containing 2 mg/mL collagenase IV (Gibco, Waltham, MA, USA) was injected into the pancreas. The pancreas was subsequently removed, and islets were collected by centrifugation, and handpicked three times under a stereomicroscope until a population of pure islets was obtained. Approximately 100 islets per mouse were obtained.

## Quantitative real-time PCR

Total RNA was extracted from samples using TRIzol (Invitrogen, Waltham, MA, USA) and an RNeasy kit (Qiagen, Hilden, Germany). Complementary DNA was synthesized from 200-ng total RNA using a ReverTra Ace qPCR RT Kit (Toyobo, Osaka, Japan). Quantitative real-time PCR (qRT–PCR) was carried out using a Brilliant III Ultra-Fast SYBR Green QPCR Master Mix (Agilent Technologies, Santa Clara, CA, USA), and samples were run using Stratagene Mx3000p. The relative mRNA levels of phosphoenolpyruvate carboxykinase (PEPCK) and glucose-6-phosphate (G6PC) in liver, and tumor necrosis factor (TNF), F4/80, and Cd11c in epididymal WAT, were assessed by qRT–PCR using GAPDH as an internal control. Ki67, CyclinA2, CyclinB1, Foxm1, CyclinD1, CyclinD2, Ins1 or Ins2 mRNAs in islets were assessed using β-actin as an internal control. The qRT–PCR reactions were carried out, and relative

mRNA expression levels were calculated using a comparative Ct method as described previously [19]. The primer sequences are listed in S1 Table.

## Immunohistochemistry

Pancreatic tissue samples were fixed with 4% paraformaldehyde at 4˚C and then embedded in paraffin. Serial sections of 4-μm thickness were cut off from each paraffin block at 200-μm intervals, and five sections were randomly selected per mouse and deparaffinized. The sections were incubated with guinea pig anti-insulin polyclonal antibody (1:100; ab7842; Abcam, Cambridge, UK), or rabbit anti-Ki67 monoclonal antibody (1:100; GTX16667; GeneTex, Irvine, CA, USA) for 48 h at 4˚C. The sections were then treated with Alexa Fluor 488-conjugated goat anti-guinea pig IgG secondary antibody (1:500; A11073; Invitrogen) or Alexa Fluor 568-conjugated goat anti-rabbit IgG secondary antibody (1:500; A11036; Invitrogen) for 1.5 h at room temperature. The total areas of insulin-positive cells and the number of islets were analyzed, and Ki67-positive β-cells were quantitatively assessed as a percentage of the total number of β-cells. All fluorescently stained sections were examined with a BZ9000 fluorescent microscope system (Keyence, Osaka, Japan).

## Histological analysis

Liver and epididymal WAT were fixed with 4% paraformaldehyde at 4˚C, embedded in paraffin, sectioned at 4 μm and then stained with hematoxylin and eosin. Soleus muscle was frozen in liquid nitrogen-cooled isopentane and cut into 10-μm thick cross sections on a cryostat at −20˚C. Sections were stained with Oil Red O (ab150678; Abcam), a marker of intramyocellular lipid content.

## Statistical analyses

The statistical significance of the differences between groups was analyzed by either unpaired *t*-tests or by one-way ANOVA with repeated measures followed by Tukey's test using SPSS statistic 26 (IBM, Endicott, NY, USA). Results are expressed as mean ± standard error of the mean (SEM). Differences were considered statistically significant at $P < 0.05$.

# Results

## Body weight gain and liver weight were decreased in HFD plus NaCl-fed mice

During 30 weeks of experiments, there were no unexpected deaths of mice in any group. After 30 weeks of feeding, significant differences in BW (NCD = 32.9 ± 0.7 and NCD plus NaCl = 32.0 ± 0.4 g) and food intake between NCD-fed and NCD plus NaCl-fed mice were not found (S1A and S1B Fig). The weights of WAT, liver, and muscle were also not significantly different between the two groups (S1C–S1H Fig). In contrast, BW was significantly lower in HFD plus NaCl-fed mice compared to that in HFD-fed mice after 30 weeks of feeding (HFD = 54.9 ± 0.5 and HFD plus NaCl = 49.1 ± 0.8 g; Fig 1A). Food intake was also significantly lower in HFD plus NaCl-fed mice during 1 week of feeding (Fig 1B). Feed efficiency (ΔBW/Δfood intake) was significantly lower in HFD plus NaCl-fed mice compared to that in HFD-fed mice (Fig 1C), but no differences between HFD-fed and HFD plus NaCl-fed mice were observed for VO$_2$ or locomotor activity (S2A and S2B Fig). Compared to HFD-fed mice, HFD plus NaCl-fed mice showed a lower liver (Fig 1D) and higher epididymal fat pad (Fig 1E) weights. Differences were not found in mesenteric fat pad (Fig 1F), subcutaneous fat pad (Fig 1G), and muscle (Fig 1H–1I) weights between these two groups. Histological analysis of liver

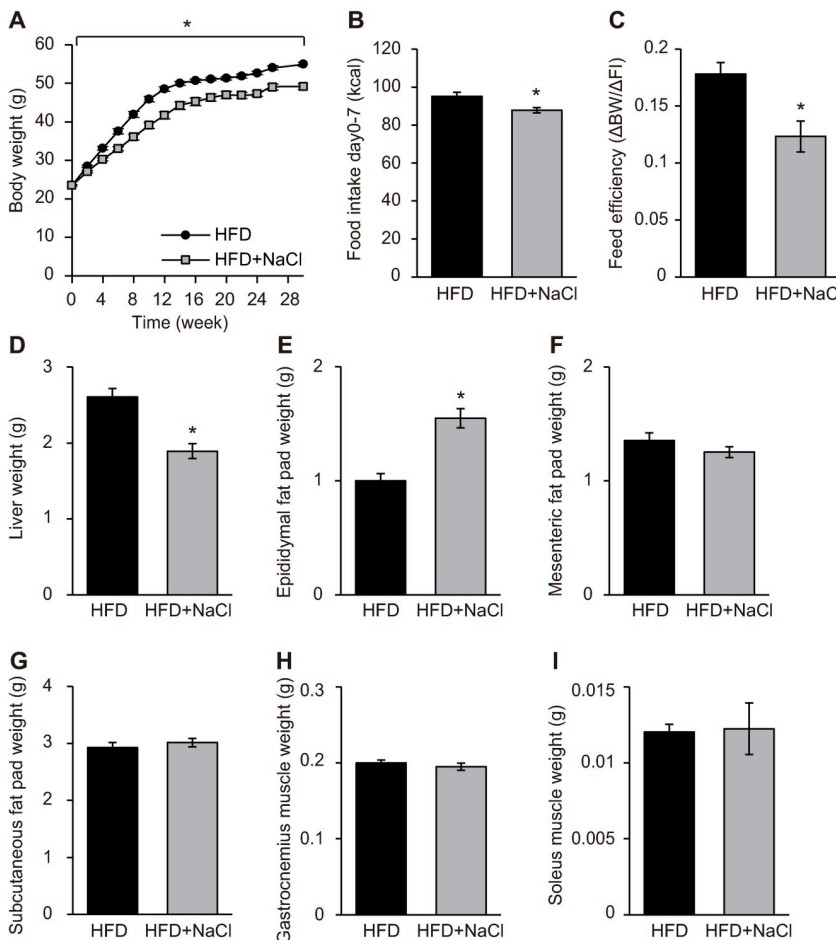

**Fig 1. Excessive NaCl intake decreased body weight gain and liver weight under HFD conditions.** (A) Body weights in mice fed a high-fat diet (HFD) or HFD plus NaCl for 30 weeks (n = 28/group). (B) Food intake in mice fed a HFD or HFD plus NaCl for 1 week (n = 15–16/group). (C) Feed efficiency in mice fed a HFD or HFD plus NaCl for 3 days (n = 10–12/group). (D) Liver, (E) epididymal fat pad, (F) mesenteric fat pad, (G) subcutaneous fat pad, (H) gastrocnemius muscle and (I) soleus muscle weights in mice fed a HFD or HFD plus NaCl for 30 weeks (n = 12–16/group). All values are mean ± SEM. $^{*}p < 0.05$ versus mice fed a HFD.

revealed reduced liver lipid deposition in HFD plus NaCl-fed mice compared to HFD-fed mice (S3A Fig). The size of adipocytes in epididymal WAT and the expression level of TNF, F4/80, and Cd11c mRNAs did not differ between these two groups (S3B and S3C Fig). Decreased Oil Red O staining, a marker of intramyocellular lipid in the soleus muscle, was observed in HFD plus NaCl-fed mice compared to HFD-fed mice (S3D Fig). Serum total cholesterol and triglyceride levels were significantly higher in HFD-fed and HFD plus NaCl-fed mice than in NCD-fed mice (S2 Table). Compared to HFD-fed mice, however, HFD plus NaCl-fed mice had lower serum total cholesterol levels; significant differences in serum free fatty acid levels between groups were not observed (S2 Table).

## HFD plus NaCl–fed mice had higher plasma glucose levels and lower plasma insulin levels in a GTT

The ITT and GTT showed no differences in blood glucose levels between NCD-fed and NCD plus NaCl-fed mice after 25 and 26 weeks of feeding, respectively (S4A and S4B Fig). However,

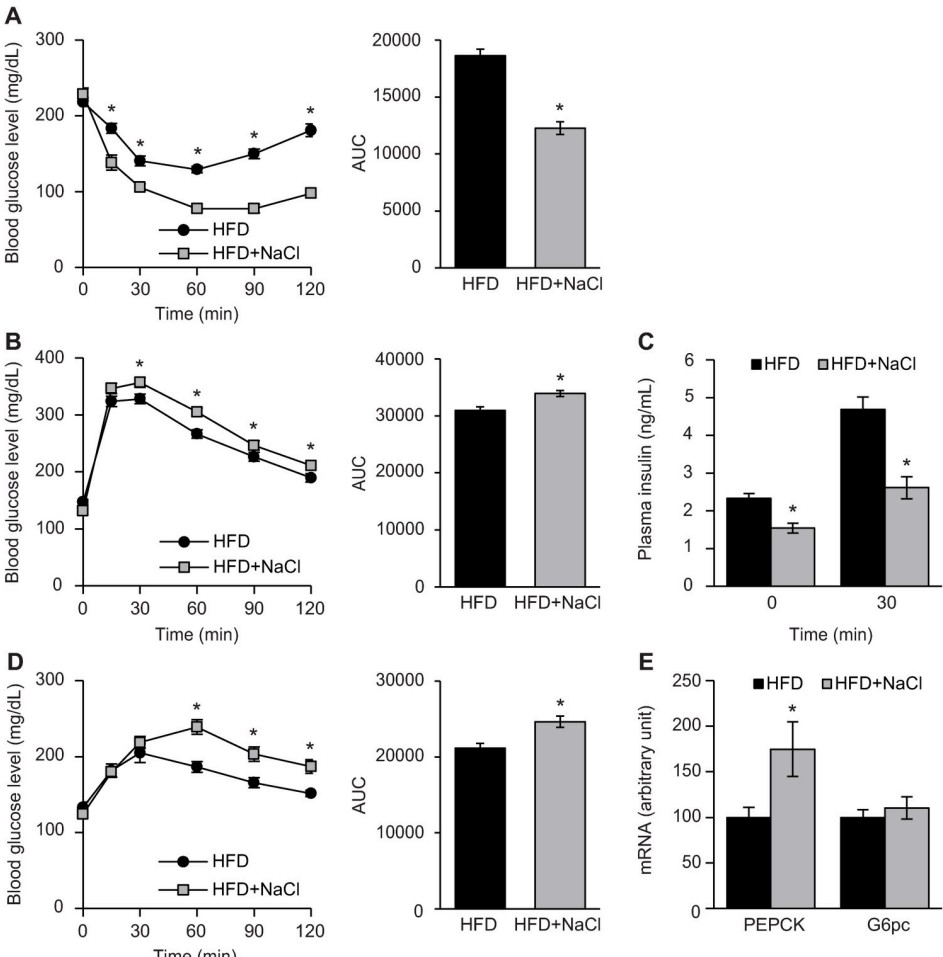

**Fig 2. Excessive NaCl intake resulted in glucose intolerance under HFD conditions.** (A) The insulin tolerance test (ITT) and area under the curve (AUC) for mice fed a high-fat diet (HFD) or HFD plus NaCl for 25 weeks (n = 11–12/group). (B) Glucose tolerance test (GTT) and the AUC for mice fed a HFD or HFD plus NaCl for 26 weeks (n = 30–36/group). (C) Plasma insulin levels at 0 and 30 min after glucose injection during a GTT (n = 14–20/group). (D) The pyruvate tolerance test (PTT) and AUC for mice fed a HFD or HFD plus NaCl for 27 weeks (n = 11–15/group). (E) The mRNA expression levels of phosphoenolpyruvate carboxykinase (PEPCK) and glucose-6-phosphate (G6PC) in the liver of mice fed a HFD or HFD plus NaCl for 30 weeks (n = 11–16/group). All values are mean ± SEM. $^*p < 0.05$ versus mice fed a HFD.

HFD plus NaCl-fed mice had lower plasma glucose levels and AUC than HFD-fed mice in the ITT (Fig 2A). In addition, the GTT showed higher plasma glucose levels and AUC in HFD plus NaCl-fed mice than in HFD-fed mice (Fig 2B). Plasma insulin levels at 0 and 30 min after glucose injections were lower in HFD plus NaCl-fed mice compared to those in HFD-fed mice (Fig 2C). The PTT showed higher plasma glucose levels and AUC in HFD plus NaCl-fed mice than in HFD-fed mice (Fig 2D). The expression levels of PEPCK mRNA in the fasted state were higher in HFD plus NaCl-fed mice than in HFD-fed mice (Fig 2E).

## β-cell area and islet number were decreased in HFD plus NaCl-fed mice 30 weeks after diet intervention

Neither the β-cell area nor the number of islets were significantly different between NCD-fed and NCD plus NaCl-fed mice (S5A–S5C Fig). In comparison, both the β-cell area and islet

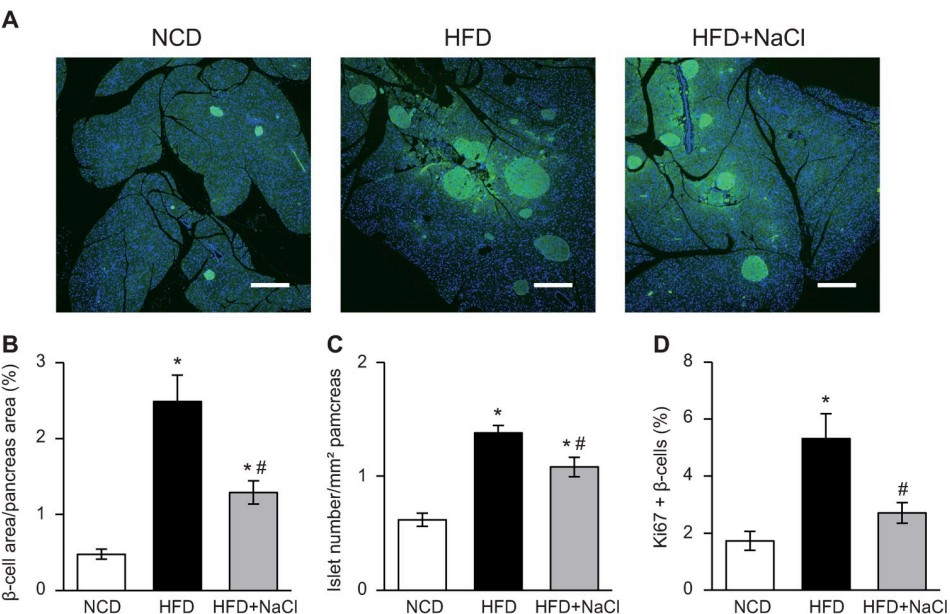

**Fig 3. Excessive NaCl intake attenuated β-cell mass expansion and proliferation under HFD conditions.** (A) Pancreatic sections stained with antibody to insulin (green) and DAPI (blue) in mice fed a normal chow diet (NCD), high-fat diet (HFD) or HFD plus NaCl for 30 weeks. Scale bars, 100 μm. (B) Beta-cell area relative to pancreas area and (C) the number of islets relative to pancreas area in mice fed a NCD, HFD or HFD plus NaCl for 30 weeks (n = 6–11/group). (D) The percentage of Ki67-positive β-cells in mice fed a NCD, HFD, or HFD plus NaCl for 3 days (n = 6–9/group). All values are mean ± SEM. $^*p < 0.05$ versus mice fed a NCD. $^#p < 0.05$ versus mice fed a HFD.

number were increased in HFD-fed and HFD plus NaCl-fed mice compared to NCD-fed mice. Additionally, compared to HFD-fed mice, HFD plus NaCl-fed mice showed a decrease in β-cell area and islet number after 30 weeks of feeding (Fig 3A–3C). The percentage of Ki67/insulin double-positive cells was increased in HFD-fed mice 3 days after diet intervention. In contrast, compared to HFD-fed mice, HFD plus NaCl-fed mice showed a decrease in the percentage of Ki67/insulin double-positive cells (Fig 3D).

## Ki67, CyclinB1, and CyclinD1 mRNA expression in islets were not increased in HFD plus NaCl-fed mice 1 week after diet intervention

Compared to NCD-fed mice, the expression level of Ki67 mRNA was increased in HFD-fed mice. In contrast, no significant differences in Ki67 mRNA levels were observed between HFD plus NaCl-fed and NCD-fed mice, while HFD plus NaCl-fed mice showed lower Ki67 mRNA expression levels than HFD-fed mice (Fig 4A). HFD-fed mice showed higher mRNA expression levels of CyclinB1 and CyclinD1 compared to NCD-fed mice; however, no significant differences were observed between HFD plus NaCl-fed and NCD-fed mice (Fig 4B–4F). Neither Ins1 nor Ins2 mRNA expression levels related to insulin synthesis differed between HFD-fed and HFD plus NaCl-fed mice (Fig 4G and 4H).

## Discussion

In the present study, we demonstrated that HFD plus NaCl-fed mice had higher plasma glucose and lower plasma insulin levels in the GTT than HFD-fed mice, even though HFD plus NaCl-fed mice exhibited a decrease in body weight gain and an improvement in insulin sensitivity compared to HFD-fed mice. We also demonstrated that the number of pancreatic islets

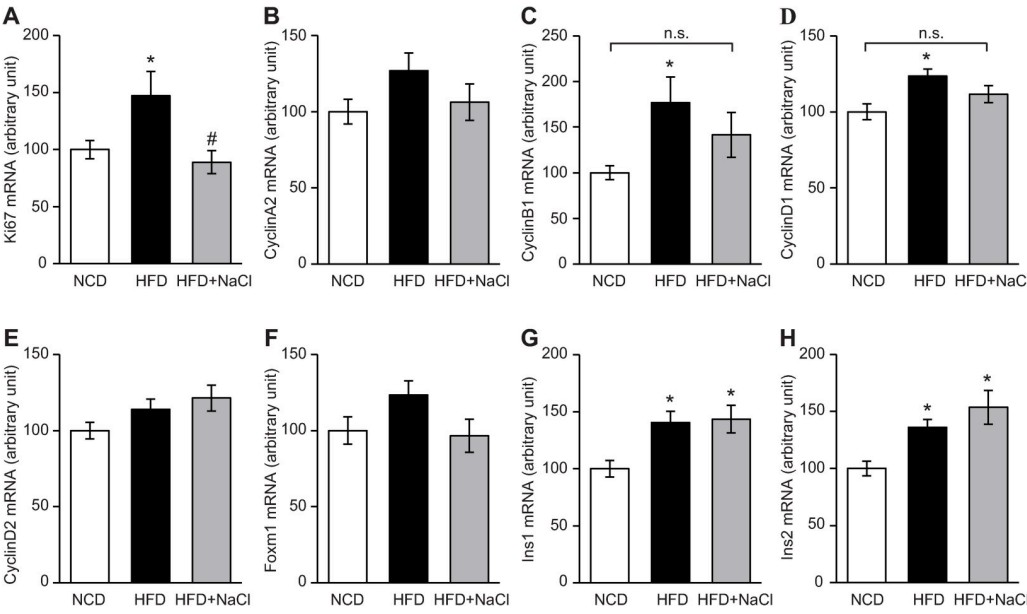

**Fig 4. Excessive NaCl intake attenuated proliferative gene expression in islets under HFD conditions.** The mRNA expression levels of (A) Ki67, (B) CyclinA2, (C) CyclinB1, (D) CyclinD1, (E) CyclinD2, (F) Foxm1, (G) Ins1, and (H) Ins2 in islets isolated from mice fed a normal chow diet (NCD), high-fat diet (HFD) or HFD plus NaCl for 1 week (n = 12–21/ group). All values are mean ± SEM. $^*p < 0.05$ versus mice fed a NCD. $^\#p < 0.05$ versus mice fed a HFD. n.s., not significant.

and the β-cell area were decreased in HFD plus NaCl-fed mice compared to HFD-fed mice. An attenuation of the HFD-induced increase in Ki67/insulin double-positive cells, Ki67, CyclinB1, and CyclinD1 mRNA expression was found in HFD plus NaCl-fed mice. Comparing NCD-fed to NCD plus NaCl-fed mice, we did not observe differences in BW gain, body composition, insulin sensitivity, glucose metabolism, β-cell area or islet number between these two groups.

Consistent with previous studies [20,21], the BW gain observed in HFD plus NaCl-fed mice was lower than that in HFD-fed mice. We also found that food intake and feed efficiency in HFD plus NaCl-fed mice were decreased compared to in HFD-fed mice; however, energy expenditure was not significantly different between these two groups. In a previous study, dietary NaCl content was found to suppress digestive efficiency and BW gain during HFD feeding [20]. It has been suggested that not only a decrease in food intake in the early phase of a diet intervention but also the suppression of digestive efficiency might lead to lower BW gain in HFD plus NaCl-fed mice.

The association between excessive NaCl consumption and insulin sensitivity has been controversial. One study using hyperinsulinemic euglycemic clamps in rats showed that excess NaCl consumption impaired insulin sensitivity; however, other studies have shown no effect on insulin sensitivity [12–14]. In our study, C57 BL/6J mice fed a HFD plus NaCl showed an improvement in insulin sensitivity. Ectopic fat deposition in non-adipose tissue, such as in the liver and soleus muscle, is strongly associated with the development of insulin resistance [22,23]. We observed the decreased accumulation of fat in the liver and soleus muscle. An improvement in insulin sensitivity observed in HFD plus NaCl-fed mice might be mediated via a decrease in fat deposition in the liver and muscle as observed in our histological analysis. Our data also showed that serum total cholesterol levels were decreased in HFD plus NaCl-fed mice compared to HFD-fed mice, which might be related to the decreased accumulation of fat

in the livers of HFD plus NaCl-fed mice [24]. We showed that HFD plus NaCl-fed mice exhibited an increase in the weight of epididymal WAT compared to HFD-fed mice, while expression levels of TNF, F4/80, and Cd11c mRNAs were similar between HFD-fed and HFD plus NaCl-fed mice. A previous study also showed that HFD plus NaCl-fed mice exhibited an increase in the weight of epididymal fat pads, but had similar levels of inflammatory cytokines in white adipose tissue and serum levels of inflammatory cytokines compared to HFD-fed mice [21]. It has been reported that macrophage-related inflammation in WAT contributes to insulin resistance [25]. From these results, an increase in the weight of epididymal fat pads in HFD plus NaCl-fed mice might not contribute to systemic insulin resistance.

HFD feeding can result in obesity, insulin resistance, glucose intolerance, and fasting hyperglycemia. An expansion of β-cells is one mechanism by which obese individuals may achieve adequate insulin secretion to meet the demands imposed by insulin resistance. Previous studies have described how the β-cell mass in HFD-fed mice was significantly increased compared to that in NCD-fed mice after 1 week of a diet intervention [26,27]. In the present study, feeding a HFD increased β-cell mass in mice compared to a NCD, while HFD plus NaCl-fed mice showed attenuated β-cell mass expansion compared to HFD-fed mice. In addition, HFD plus NaCl-fed mice showed lower plasma insulin levels than HFD-fed mice and glucose intolerance during a GTT. It is estimated that when the adaptive expansion of β-cells becomes abnormal, insufficient insulin production results in glucose intolerance. Our study suggested that excess NaCl intake accompanied by a HFD attenuated an increase in β-cell mass and resulted in a decrease in insulin secretion and glucose intolerance. Furthermore, in the present study, the feeding of a HFD plus NaCl to mice caused an increase of hepatic glucogenesis in the PTT. Insulin is a dominant suppressor of glucogenesis, and regulates hepatic glucogenesis by the suppression of PEPCK mRNA expression through the interaction of the forkhead box protein O1–peroxisome proliferator-activated receptor gamma coactivator 1α pathway [28]. It is suggested that a decrease in plasma insulin levels might cause PEPCK mRNA expression and result in hepatic glucogenesis in HFD plus NaCl-fed mice. It is plausible that an excessive intake of NaCl accompanied by a HFD improves insulin sensitivity, while the impairment of insulin secretion results in glucose intolerance.

Previous studies have shown that HFD-induced β-cell proliferation was detected after only 3 days of HFD feeding, before insulin resistance occurred [29]. Our data also showed that HFD induced β-cell proliferation after only 3 days of intervention. However, in the present study, NaCl accompanied by a HFD attenuated the HFD-induced increase of Ki67/insulin double positive β-cells and HFD-induced Ki67, CyclinB1 or CyclinD1 mRNA expression in islets. CyclinB1 has a role in promoting the transition of the G2 phase of the cell cycle to mitosis [30], and in increasing the expression level of CyclinB1 mRNA as found in islets from mice fed a HFD for 1 week [26,29,31]. In addition, previous studies outlined how β-cell proliferation was induced by cyclinD1 overexpression and an increase of CyclinD1 mRNA expression levels in HFD-fed mice [26,32]. Although a prior study described how high concentrations of NaCl in media induced the arrest of cell-cycle progression [33], the precise mechanism has not been identified. From our study, it is not clear how the intake of excessive NaCl impaired cell-cycle progression induced by HFD in islets, but it is possible that excessive NaCl feeding might suppress β-cell-proliferative gene expression and cell-cycle progression early in the HFD feeding. As a result, the attenuation of an increase in β-cell mass might occur. Furthermore, insulin and factors downstream of insulin signaling have been reported to participate in β-cell proliferation [34–36]. It may be that a decrease in plasma insulin levels observed in HFD plus NaCl-fed mice compared to HFD-fed mice might also contribute to the attenuation of β-cell proliferation. The mechanism of how a HFD plus NaCl induces the attenuation of β-cell proliferation and the expansion of β-cell mass needs to be investigated in future.

In conclusion, our data showed that excessive sodium chloride intake was associated with glucose intolerance under HFD conditions by attenuating HFD-induced β-cell proliferation and β-cell mass expansion with the impairment of insulin secretion.

## Supporting information

**S1 Fig. Body weight changes and food intake in NCD-fed or NCD plus NaCl-fed mice.** (A) Body weight and (B) cumulative food intake in mice fed a normal chow diet (NCD) or a NCD plus NaCl for 30 weeks (n = 16/group). (C) Liver, (D) epididymal fat pad, (E) mesenteric fat pad, (F) subcutaneous fat pad, (G) gastrocnemius muscle, and (H) soleus muscle weights in mice fed a NCD or a NCD plus NaCl for 30 weeks (n = 16/group). All values are mean ± SEM.
(TIF)

**S2 Fig. $VO_2$ and locomotor activity of mice fed a HFD or HFD plus NaCl.** (A) Oxygen consumption ($VO_2$) and (B) locomotor activity in mice fed a high-fat diet (HFD) or a HFD plus NaCl (n = 5/group). All values are mean ± SEM.
(TIF)

**S3 Fig. Liver, epididymal WAT, and muscle from mice fed a HFD or HFD plus NaCl.** (A) Hematoxylin and eosin staining of liver sections from mice fed a high-fat diet (HFD) or HFD plus NaCl for 30 weeks. Scale bars, 100 μm. (B) Hematoxylin and eosin staining of epididymal white adipose tissue (WAT) in mice fed a HFD or HFD plus NaCl for 30 weeks. Scale bars, 100 μm. (C) Messenger RNA expression levels of TNF, F4/80, and Cd11c in mice fed a HFD or a HFD plus NaCl for 30 weeks as assessed by quantitative real-time PCR (n = 11–16/group). (D) Oil Red O staining of soleus from mice fed a HFD or HFD plus NaCl for 30 weeks. Scale bars, 50 μm. All values are mean ± SEM.
(TIF)

**S4 Fig. Glucose metabolism of mice fed a NCD or NCD plus NaCl.** (A) An insulin tolerance test (ITT) and the area under curve (AUC) in mice fed a normal chow diet (NCD) or NCD plus NaCl for 25 weeks (n = 14–15/group). (B) A glucose tolerance test (GTT) and AUC in mice fed a NCD or NCD plus NaCl for 26 weeks (n = 13–16/group). All values are mean ± SEM.
(TIF)

**S5 Fig. Analysis of pancreas from mice fed a NCD or NCD plus NaCl.** (A) Pancreatic sections stained with antibody to insulin (green) and DAPI (blue) in mice fed a normal chow diet (NCD) or NCD plus NaCl for 30 weeks. Scale bars, 100 μm. (B) Beta-cell area relative to pancreas area, and (C) the number of islets relative to pancreas area in mice fed a NCD or NCD plus NaCl for 30 weeks (n = 6/group). All values are mean ± SEM.
(TIF)

**S1 Table. Primers used for quantitative real-time PCR.**
(TIF)

**S2 Table. Serum total cholesterol, triglyceride, and free fatty acid levels.** Serum total cholesterol, triglyceride and free fatty acid levels in mice fed a normal chow diet (NCD), high-fat diet (HFD) or HFD plus NaCl (n = 8/group). All values are mean ± SEM. $^*p < 0.05$ versus mice fed a NCD. $^\#p < 0.05$ versus mice fed a HFD.
(TIF)

## Acknowledgments

We thank Kinji Ohno for his help with the precise isolation of muscle and Atsushi Enomoto for his help with the immunohistochemistry of pancreas samples. We also thank Michiko Yamada for helpful technical assistance.

## Author Contributions

**Conceptualization:** Keigo Taki, Hiroshi Takagi, Ryoichi Banno, Hiroshi Arima.

**Data curation:** Keigo Taki.

**Formal analysis:** Keigo Taki.

**Funding acquisition:** Hiroshi Takagi.

**Investigation:** Keigo Taki, Hiroshi Takagi, Tomonori Hirose, Runan Sun, Hiroshi Yaginuma, Akira Mizoguchi, Mariko Sugiyama, Taku Tsunekawa, Yoshihiro Ito, Ryoichi Banno.

**Methodology:** Keigo Taki, Hiroshi Takagi, Mariko Sugiyama, Taku Tsunekawa, Yoshihiro Ito, Ryoichi Banno, Daisuke Sakano, Shoen Kume, Hiroshi Arima.

**Project administration:** Hiroshi Takagi, Tomoko Kobayashi, Mariko Sugiyama, Taku Tsunekawa, Takeshi Onoue, Daisuke Hagiwara, Yoshihiro Ito, Shintaro Iwama, Hidetaka Suga, Ryoichi Banno, Hiroshi Arima.

**Resources:** Ryoichi Banno, Daisuke Sakano, Shoen Kume, Hiroshi Arima.

**Supervision:** Hiroshi Takagi, Tomoko Kobayashi, Mariko Sugiyama, Taku Tsunekawa, Takeshi Onoue, Daisuke Hagiwara, Yoshihiro Ito, Shintaro Iwama, Hidetaka Suga, Ryoichi Banno, Daisuke Sakano, Shoen Kume, Hiroshi Arima.

**Validation:** Keigo Taki, Hiroshi Takagi.

**Visualization:** Keigo Taki.

**Writing – original draft:** Keigo Taki, Hiroshi Takagi.

**Writing – review & editing:** Hiroshi Arima.

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
