## [Decision Letter · Decision Letter 0]

30 Dec 2020

PONE-D-20-36592

Dietary sodium chloride attenuates increased β-cell mass to cause glucose intolerance in mice under a high-fat diet

PLOS ONE

Dear Dr. Takagi,

Thank you for submitting your manuscript to PLOS ONE. After careful consideration, we feel that it has merit but does not fully meet PLOS ONE’s publication criteria as it currently stands. Therefore, we invite you to submit a revised version of the manuscript that addresses the points raised during the review process.

We look forward to receiving your revised manuscript.

Kind regards,

Nobuyuki Takahashi, Ph.D.

Academic Editor

PLOS ONE

2. To comply with PLOS ONE submissions requirements, in your Methods section, please provide additional information regarding the experiments involving animals and ensure you have included details on (1) methods of sacrifice, (2) methods of anesthesia and/or analgesia, and (3) efforts to alleviate suffering. Moreover, please clarify how many animals were used, and how they were assigned to the different experimental groups.

Reviewers' comments:

Reviewer's Responses to Questions

**Comments to the Author**

1. Is the manuscript technically sound, and do the data support the conclusions?

Reviewer #1: Yes

Reviewer #2: Yes

2. Has the statistical analysis been performed appropriately and rigorously? 

Reviewer #1: Yes

Reviewer #2: I Don't Know

3. Have the authors made all data underlying the findings in their manuscript fully available?

Reviewer #1: Yes

Reviewer #2: Yes

4. Is the manuscript presented in an intelligible fashion and written in standard English?

Reviewer #1: Yes

Reviewer #2: Yes

5. Review Comments to the Author

Reviewer #1: This manuscript has an important finding and scientific merit. Although this manuscript is well written, I have some concerns:

1) Are there four figures? The figure numbers are confusing. The manuscript shows four figures in the text but there are more and more attached figures. Also, there is a lot of supporting S figures.

2) The Area Under the Curve (AUC) is derived from the Oral Glucose Tolerance Test (OGTT) which is widely used to diagnose the Impaired Glucose Tolerance (IGT) in the clinic. This expression is not explained in the manuscript. The journal is interdisciplinary and this term should be explained to readers.

3) What is the mortality rate of animals? Why mice but nor rats?

4) It is recommended that the expression "HFD + NaCl-fed mice" is replaced by "HFD & NaCl-fed mice" or "HFD plus NaCl-fed mice" in the text.

However, the discussion is impressive and the conclusion is clear.

Reviewer #2: Authors demonstrated here that a combination of HF and high NaCl caused attenuation of increase in beta-cell mass, resulting in glucose intolerance. These findings are very interesting and valuable. Experimental design is also excellent.

However, authors should show blood parameters such as serum levels of free fatty acids and TG in addition to glucose. This is because these parameters are important as basic information of animal experiments in metabolism research.

6. PLOS authors have the option to publish the peer review history of their article (what does this mean?). If published, this will include your full peer review and any attached files.

Reviewer #1: **Yes: **Mahmoud Esmat Balbaa

Reviewer #2: No

---

## [Author Response · Author response to Decision Letter 0]

12 Feb 2021

Reviewer #1: This manuscript has an important finding and scientific merit. Although this manuscript is well written, I have some concerns:

1) Are there four figures? The figure numbers are confusing. The manuscript shows four figures in the text but there are more and more attached figures. Also, there is a lot of supporting S figures.

There are four figures and five supplemental figures. We have indicated the figure number on each figure and submitted supplemental figures and tables as Supporting Information.. 

2) The Area Under the Curve (AUC) is derived from the Oral Glucose Tolerance Test (OGTT) which is widely used to diagnose the Impaired Glucose Tolerance (IGT) in the clinic. This expression is not explained in the manuscript. The journal is interdisciplinary and this term should be explained to readers.

Based on the comment, we have added an explanation in the revised manuscript (page 6, lines 105–106 and 110–111). 

3) What is the mortality rate of animals? Why mice but nor rats?

We did not observe any unexpected deaths of mice during the experiments, and have commented on this point in the revised manuscript (page 10, line 179). Please understand that it is difficult to perform experiments with rats as space within our animal facility is limited. 

4) It is recommended that the expression "HFD + NaCl-fed mice" is replaced by "HFD & NaCl-fed mice" or "HFD plus NaCl-fed mice" in the text.

However, the discussion is impressive and the conclusion is clear. 

According to the advice given, we have changed the expression “HFD + NaCl-fed” to “HFD plus NaCl-fed mice” in the revised manuscript. 

Reviewer #2: Authors demonstrated here that a combination of HF and high NaCl caused attenuation of increase in beta-cell mass, resulting in glucose intolerance. These findings are very interesting and valuable. Experimental design is also excellent.

However, authors should show blood parameters such as serum levels of free fatty acids and TG in addition to glucose. This is because these parameters are important as basic information of animal experiments in metabolism research.

Based on the advice given, we present data on serum levels of total cholesterol, triglycerides and free fatty acid, as well as an interpretation of results in the revised manuscript (page 11, lines 179–201, page 17, lines 316–319, and S2 Table).

---

## [Editor Report · Decision Letter 1]

19 Feb 2021

Dietary sodium chloride attenuates increased β-cell mass to cause glucose intolerance in mice under a high-fat diet

PONE-D-20-36592R1

Dear Dr. Takagi,

We’re pleased to inform you that your manuscript has been judged scientifically suitable for publication and will be formally accepted for publication once it meets all outstanding technical requirements.

Kind regards,

Nobuyuki Takahashi, Ph.D.

Academic Editor

PLOS ONE

Additional Editor Comments:

Authors properly addressed all of the Reviewers' comments.

---

## [Editor Report · Acceptance letter]

23 Feb 2021

PONE-D-20-36592R1 

Dietary sodium chloride attenuates increased β-cell mass to cause glucose intolerance in mice under a high-fat diet 

Dear Dr. Takagi:

I'm pleased to inform you that your manuscript has been deemed suitable for publication in PLOS ONE. Congratulations! Your manuscript is now with our production department. 

Kind regards, 

on behalf of

Dr. Nobuyuki Takahashi 

Academic Editor

PLOS ONE